# Razuprotafib Does Not Improve Microcirculatory Perfusion Disturbances nor Renal Edema in Rats on Extracorporeal Circulation

**DOI:** 10.3390/ijms26073000

**Published:** 2025-03-25

**Authors:** Dionne P. C. Dubelaar, Carolien Volleman, Philippa G. Phelp, Roselique Ibelings, Iris Voorn, Anita M. Tuip-de Boer, Chantal A. Polet, Joris J. Roelofs, Alexander P. J. Vlaar, Matijs van Meurs, Charissa E. van den Brom

**Affiliations:** 1Department of Intensive Care Medicine, Amsterdam UMC, University of Amsterdam, 1105 AZ Amsterdam, The Netherlands; d.p.c.dubelaar@amsterdamumc.nl (D.P.C.D.); c.volleman@amsterdamumc.nl (C.V.); p.phelp@amsterdamumc.nl (P.G.P.); a.m.deboer@amsterdamumc.nl (A.M.T.-d.B.); c.a.polet@amsterdamumc.nl (C.A.P.); a.p.vlaar@amsterdamumc.nl (A.P.J.V.); 2Laboratory for Experimental Intensive Care and Anesthesiology (LEICA), Amsterdam UMC, University of Amsterdam, 1105 AZ Amsterdam, The Netherlands; 3Department of Anesthesiology, Amsterdam UMC, VU University, 1081 HV Amsterdam, The Netherlands; 4Department of Pathology, Amsterdam UMC, University of Amsterdam, 1105 AZ Amsterdam, The Netherlands; j.j.roelofs@amsterdamumc.nl; 5Department of Critical Care, University Medical Center Groningen, 9713 GZ Groningen, The Netherlands; m.van.meurs@umcg.nl

**Keywords:** ECMO, edema, endothelium, extracorporeal circulation, microcirculation, perfusion, permeability, Tie2, vascular dysfunction, vascular leakage, VE-PTP

## Abstract

Extracorporeal membrane oxygenation (ECMO) can be a life-saving intervention, but it is associated with high complication rates. ECMO induces systemic inflammation and endothelial hyperpermeability, thereby causing tissue edema, microcirculatory perfusion disturbances, and organ failure. This study investigated whether the inhibition of vascular endothelial protein tyrosine phosphatase (VE-PTP), a regulator of endothelial permeability, reduces extracorporeal circulation (ECC)-induced microvascular dysfunction. Rats were subjected to ECC after treatment with Razuprotafib (n = 11) or a placebo (n = 11), or they underwent a sham procedure (n = 8). Razuprotafib had no effect on the ECC-induced impairment of capillary perfusion, as assessed with intravital microscopy, nor did it influence the increased wet-to-dry weight ratio in kidneys, a marker of edema associated with ECC. Interestingly, Razuprotafib suppressed the ECC-induced increase in TNFα, whereas angiopoietin-2 even further increased, following the discontinuation of ECC. Circulating interleukin-6, ICAM-1, angiopoietin-1, and soluble Tie2 and tissue *VE-PTP*, *Tie1*, and *Tie2* mRNA expression were not affected by Razuprotafib. Furthermore, Razuprotafib improved the PaO_2_/FiO_2_ ratio and reduced histopathological pulmonary interstitial inflammation following ECC compared to the placebo. To conclude, treatment with Razuprotafib did not improve ECC-induced microcirculatory perfusion disturbances nor renal edema.

## 1. Introduction

Extracorporeal membrane oxygenation (ECMO) is a treatment modality that provides temporary life support for critically ill patients with severe cardiac and/or respiratory failure. The use of ECMO therapy has doubled from 2010 to 2020 [1], and although patient outcomes have improved, morbidity and mortality are still unacceptably high [2]. Unfortunately, evidence-based treatment strategies to prevent or treat organ failure in patients on ECMO support are scarce, highlighting the critical need for further research to improve patient outcome.

A crucial component of ECMO is the extracorporeal circuit and thereby extracorporeal circulation (ECC), which leads to contact activation. As blood comes in contact with non-biological surfaces, it triggers an inflammatory response [3]. This inflammatory response not only initiates coagulation, but it also activates the endothelium to promote the expression of adhesion molecules, a process related to organ failure in patients on veno-arterial (VA) ECMO support [4,5]. Endothelial activation can result in a hyperpermeable endothelium with leakage of fluid to the interstitium, subsequently leading to tissue edema, leukocyte recruitment into organs, microcirculatory perfusion disturbances, and as a result, organ failure.

The pathogenesis of ECC-induced microvascular complications is multifactorial, but evidence supports a vital role for endothelial hyperpermeability. Previously, we and others have shown that ECC in rats resulted in endothelial damage, vascular leakage, tissue edema, and microcirculatory perfusion disturbances [6,7,8,9,10]. One of the major regulators of endothelial permeability is the endothelial angiopoietin/Tie2 system [11]. This pathway consists of three ligands, angiopoietins-1, -2, and -3/4, whose activities are mediated through the tyrosine kinase receptors Tie1 and Tie2 and vascular endothelial protein tyrosine phosphatase (VE-PTP). The main receptor Tie2 is located almost exclusively on endothelial cells and has to be dimerized and phosphorylated to be active. During quiescence, angiopoietin-1 activates Tie2, ensuring the integrity of the endothelial barrier, whereas angiopoietin-2 functions as a context-dependent agonist/antagonist of Tie2. In response to inflammation, the angiopoietin-2-mediated inhibition of Tie2 is linked to disruption of the endothelial barrier and tissue edema. Dysregulation of the endothelial angiopoietin/Tie2 system occurs in rats with ECC, with decreased tissue *Tie2* mRNA expression and increased circulating angiopoietin-2 levels [6]. Moreover, high circulating angio-poietin-2 levels during ECMO support are linked to poor patient outcomes [12,13]. Interestingly, the pharmacological activation of Tie2 with the angiopoietin-1 mimetic vasculotide prevented pulmonary vascular leakage and microcirculatory perfusion disturbances in rats with ECC [6], highlighting the promising effects of targeting the endothelial angiopoietin/Tie2 system to reduce microvascular dysfunction.

In addition to angiopoietin-1 and -2, VE-PTP is also a regulator of Tie2. The VE-PTP receptor is an endothelial-specific phosphatase, which enzymatically removes phosphates from proteins with which it associates, such as Tie2 or vascular endothelial (VE)-cadherin [14]. The association of VE-PTP with Tie2 dephosphorylates Tie2 and thereby destabilizes the endothelial barrier [15]. Interestingly, the inhibition of VE-PTP enhances Tie2 phosphorylation and strengthens the endothelial barrier against inflammatory triggers [16], which makes VE-PTP another target of interest to reduce endothelial hyperpermeability. Razuprotafib (AKB-9778) is a small molecule inhibitor of VE-PTP, and its safety has been proven for long-term treatment in patients with diabetic macular edema [17]. Moreover, Razuprotafib could counteract inflammation-induced vascular hyperpermeability in the skin of mice [16]. Whether Razuprotafib protects against microvascular dysfunction in the acute inflammatory setting has not been studied. Therefore, as a first translational step, we have investigated whether Razuprotafib reduces ECC-induced microvascular dysfunction in rats. Here, we show that treatment with Razuprotafib mitigates ECC-induced lung injury, without affecting ECC-induced organ hyperpermeability or microcirculatory perfusion disturbances.

## 2. Results

The initiation of ECC reduced heart rate, hematocrit, partial pressure of carbon dioxide, bicarbonate levels, and base excess, and it increased pH and lactate levels compared to the control rats (Figure 1). The discontinuation of ECC immediately increased mean arterial pressure compared to the controls (Figure 1A). After one hour of the discontinuation of ECC, only base excess and lactate levels were completely restored back to baseline levels (Figure 1H,I). Oxygen tension (Figure 1G)remained unchanged over time in rats with ECC compared to the controls.

Treatment with Razuprotafib resulted in a higher increase in lactate levels after the start of ECC (Figure 1I). Moreover, a higher oxygen tension was observed after the discontinuation of ECC in Razuprotafib-treated rats compared to the placebo-treated rats (Figure 1G). Other hemodynamic and blood gas parameters were not affected by Razuprotafib administration.

Interestingly, the administration of Razuprotafib induced an immediate decrease in mean arterial pressure (ΔMAP: −37 [−40–−19] mmHg), which was in contrast to vehicle treatment (ΔMAP: 11 [8–17] mmHg, *p* = 0.007). The ECC-induced decrease in MAP did not differ significantly between Razuprotafib- and vehicle-treated rats (ΔMAP ECC + razu: −42 [−48–−19] vs. ECC: −21 [−36–−10] mmHg, *p* = 0.14). 

### 2.1. ECC Disturbed Microcirculatory Perfusion

ECC immediately reduced the number of continuously perfused vessels and the proportion of perfused vessels compared to the controls (Figure 2A,D). This was paralleled by an increase in the number of non-perfused vessels, without alterations in intermittently perfused vessels (Figure 2B,C). Upon the discontinuation of ECC, there was a partial recovery in capillary perfusion; however, perfusion did not return to baseline levels or match those of the control group (Figure 2).

Razuprotafib did not affect the number of continuously, intermittently, and non-perfused vessels nor the proportion of perfused vessels compared to vehicle administration during and after ECC (Figure 2).

### 2.2. ECC Induced Edema in the Kidney, but Not in the Lung

Following ECC, an increase in the wet-to-dry weight ratio was observed in the kidneys compared to the controls (Figure 3D). Interestingly, this change was not evident in the lungs, as their wet-to-dry weight ratios remained similar to those of the control group (Figure 3B). There were no differences in dextran extravasation following ECC compared to the controls in both lungs and kidneys (Figure 3A,C). Additionally, histopathological assessment did not reveal interstitial edema in the kidneys or perivascular and alveolar edema in the lungs compared to the controls (Table 1).

Razuprotafib did not affect the wet-to-dry weight ratios, dextran extravasation, or histopathological assessment of edema in either the kidneys or the lungs compared to vehicle administration (Figure 3, Table 1).

According to protocol, rats on ECC received more albumin (0.8 [0.0–1.5] vs. 0.0 [0.0–0.0] mL, *p* < 0.05), total fluids (11.8 [11.3–13.1] vs. 0.0 [0.0–0.0], *p* < 0.01), and phenylephrine (27 ± 29 vs. 0 ± 0 µg, *p* < 0.05) compared to the control rats. Razuprotafib-treated rats received comparable volumes of albumin (1.0 [0.5–1.5] mL, *p* = 0.67), phenylephrine (40 ± 25 µg, *p* = 0.41), and total fluids (13.0 [12.3–13.4] mL, *p* = 0.50) compared to the vehicle.

### 2.3. Razuprotafib Suppressed the Increase in TNFα, but Increased Angiopoietin-2

ECC increased circulating tumor necrosis factor α (TNFα), interleukin-6 (IL-6), and intercellular adhesion molecule 1 (ICAM-1) levels compared to controls, with IL-6 and ICAM-1 levels continuing to increase after the discontinuation of ECC (Figure 4A–C). Interestingly, Razuprotafib treatment suppressed the increase in circulating TNFα during ECC compared to vehicle rats, without affecting IL-6 and ICAM-1 levels (Figure 4A–C).

Circulating angiopoietin-1 and soluble Tie2, but not angiopoietin-2, levels significantly increased during ECC compared to the controls (Figure 4D–F). The discontinuation of ECC lowered angiopoietin-1 and soluble Tie2 levels, yet circulating angiopoietin-2 increased compared to the controls. Treatment with Razuprotafib did not affect circulating angiopoietin-1 or soluble Tie2 levels, but it unexpectedly further increased circulating angiopoietin-2 levels compared to untreated rats after the discontinuation of ECC (Figure 4E).

### 2.4. Molecular Alterations in the Angiopoietin/Tie2 System

In the lung, *Tie2* and *VE-PTP* mRNA expression significantly decreased in rats on ECC compared to the control rats (Figure 5A,B); however, *VE-PTP* did not reach statistical significance (*p* = 0.058). Lung *Tie1* mRNA expression was not altered in rats on ECC (Figure 5C). Razuprotafib did not affect *Tie2* and *VE-PTP* mRNA expression compared to the vehicle in rats on ECC, but it seemed to protect against the decrease in *Tie1* mRNA expression, which also did not reach statistical significance (*p* = 0.050).

In the kidneys, mRNA expression of *Tie2*, *VE-PTP*, and *Tie1* was not affected by ECC nor Razuprotafib treatment (Figure 5D–F).

### 2.5. Razuprotafib Reduces Lung Injury Following ECC

One hour of ECC reduced the PaO_2_/FiO_2_ ratio, which is the ratio of arterial oxygen tension to fractional inspired oxygen, compared to the control rats (303 [166–675] vs. 658 [563–878], *p* < 0.05). Interestingly, 60 min after the discontinuation of ECC, rats treated with Razuprotafib had a higher PaO_2_/FiO_2_ ratio compared to untreated rats (Figure 6A).

The histopathological assessment of lung tissue showed interstitial inflammation following ECC compared to the controls, which was significantly lower in rats on ECC treated with Razuprotafib compared to the vehicle (Figure 6B). No alterations were found in neutrophil sequestration, hemorrhage, or thrombi in lungs between all three groups (Table 1). The histopathological assessment of the kidneys did not show tubular injury following ECC compared to the controls nor an effect of Razuprotafib treatment (Table 1).

## 3. Discussion

Extracorporeal membrane oxygenation (ECMO) induces systemic inflammation and endothelial hyperpermeability, thereby causing tissue edema, microcirculatory perfusion disturbances, and multiple organ dysfunction. The endothelial angiopoietin/Tie2 system is of major importance in the regulation of endothelial permeability, and the therapeutic targeting of this system is a promising method for reducing microvascular dysfunction. Here, we investigated whether the inhibition of the endothelial VE-PTP receptor using Razuprotafib reduced extracorporeal circulation (ECC)-induced microvascular dysfunction. We showed that ECC in rats induced systemic inflammation, endothelial activation, and damage. Moreover, ECC impaired respiratory function and microcirculatory perfusion, which was paralleled by renal, but not pulmonary, edema and a disturbed endothelial angiopoietin/Tie2 system. Treatment with Razuprotafib did not improve microcirculatory perfusion nor reduced renal edema. Interestingly, Razuprotafib suppressed part of the ECC-induced inflammatory response and pulmonary interstitial inflammation and improved respiratory function. However, circulating angiopoietin-2 levels further increased upon treatment with Razuprotafib. Taken together, Razuprotafib did not improve ECC-induced microcirculatory perfusion disturbances nor renal edema. Although minor, preliminary findings suggest that Razuprotafib has beneficial effects on ECC-induced lung injury, which warrants further investigation before advancing to the next phase.

Microcirculatory perfusion is essential for the tissue delivery of oxygen and nutrients, with disturbances linked to multiple organ dysfunction. We and others have shown that microcirculatory hypoperfusion persists for days in patients undergoing cardiac surgery with cardiopulmonary bypass [18,19], which is primarily attributed to the effects of ECC [20]. In critically ill patients on VA-ECMO support, data on microcirculatory perfusion are limited and inconclusive, but survivors seem to have better perfusion than non-survivors [21]. In the present study, we confirmed that ECC induces systemic inflammation and endothelial activation and damage in healthy rats. Moreover, we showed the deteriorating effects of ECC on microcirculatory perfusion, respiratory function, and kidney edema formation. In contrast to previous findings, we did not show vascular leakage or histopathological signs of edema in either the lungs or kidneys. This discrepancy may arise from the differences in techniques or the use of Ringer’s lactate and albumin as the priming strategy of the extracorporeal circuit [22]. Taken together, ECC induced systemic inflammation and endothelial activation and damage, which was paralleled by renal edema, microcirculatory perfusion disturbances, and impaired respiratory function in healthy rats.

The endothelial angiopoietin/Tie2 system is of major importance in the regulation of endothelial permeability. We found that ECC increased circulating levels of angiopoietin-2 and decreased pulmonary *Tie2* gene expression. Renal *Tie2* expression remained unaltered, which is surprising, as ECC caused renal and not pulmonary edema. Previously, we and others reported that the suppression of Tie2 expression resulted in both renal and pulmonary edema [23,24]. A limitation of the current study is that we were unable to determine the phosphorylation of Tie2 and VE-PTP in these organs and, consequently, its activation status. Despite this, the pharmacological activation of Tie2 via the angiopoietin-1 mimetic vasculotide showed promise in mitigating microvascular dysfunction in critical illness [6,25,26,27,28,29]. Notably, Tie2 regulation extended beyond angiopoietin-1, with the endothelial VE-PTP receptor as another critical regulator. Previously, it was shown that the genetic and pharmacological inhibition of VE-PTP suppresses endothelial hyperpermeability via the activation of Tie2 [16,30]. Contrary to our hypothesis, the inhibition of VE-PTP with Razuprotafib did not restore microcirculatory perfusion disturbances nor did it affect renal edema. There are several possible explanations for the absence of this effect. First, ECC did not increase *VE-PTP* gene expression in lungs and kidneys; however, mRNA levels do not provide information on the protein and phosphorylation status of VE-PTP. Surprisingly, *VE-PTP* gene expression was even reduced in lung tissue following ECC. Contrasting results have been reported on VE-PTP expression. In the kidneys, VE-PTP expression increased with diabetes, hypertension, and hypoxia [31,32], while in the lungs, it increased with diabetes [31], remained unchanged after LPS or VEGF stimulation [33], and decreased with injurious mechanical ventilation [34]. While the highest expression is in the kidney and lungs, its location in the vasculature is significant, with weak expression in peritubular capillaries and strong expression in glomerular capillaries [31].

Second, in addition to Tie2, VE-PTP also associates with VE-cadherin, an adherens junction protein important in cell–cell adhesion. VE-PTP is required for the proper function of VE-cadherin [35]. The association of VE-PTP with VE-cadherin inhibits the phosphorylation of VE-cadherin and stabilizes the endothelial barrier function [36]. VE-PTP dissociation from VE-cadherin is necessary for opening endothelial junctions and occurs upon inflammatory stimuli [33,35,37]. This highlights that the inhibition of VE-PTP has opposing effects on endothelial permeability and seems dependent on the presence of VE-cadherin and Tie2. In the absence of VE-cadherin, the blocking of VE-PTP stabilized endothelial junctions via Tie2, whereas in the absence of Tie2, the blocking of VE-PTP destabilized endothelial barrier integrity [16]. The complex interaction between these three endothelial surface proteins determines vascular permeability. In our ECC model, VE-cadherin gene expression was unaffected in both organs (unpublished data), while *Tie2* gene expression was decreased in the lungs. Moreover, Razuprotafib further increased circulating angiopoietin-2 levels. Angiopoietin-2 can be a context-dependent agonist or antagonist. During inflammation, angiopoietin-2 is originally known as an antagonist for the Tie2 receptor and, thereby, destabilization of the endothelial barrier. This is specific for the blood endothelium, as in the lymphatic vasculature angiopoietin-2 possesses an agonistic role [38]. Several conditions have previously been identified in which angiopoietin-2 acts as an agonist, such as the presence of the endothelial Tie1 receptor [39] and high concentrations of angiopoietin-2 in vitro [40], but the inhibition of VE-PTP has also been shown to convert angiopoietin-2 into a potent Tie2 activator in blood endothelium [41]. Moreover, it was recently revealed that the inflammation-induced upregulation of angiopoietin-2 leads to cleavage of full-length angiopoietin-2 to smaller fragments by cathepsin K, and it converts angiopoietin-2 from a Tie2 agonist to an antagonist [42]. Taken together, these findings suggest that the increase in angiopoietin-2 induced by Razuprotafib may have both beneficial and detrimental effects on the endothelial barrier, with the balance between these beneficial and detrimental effects being context-dependent and difficult to delineate in our model.

Interestingly, we observed minor mitigating effects of Razuprotafib on systemic inflammation, pulmonary interstitial inflammation, and respiratory function. To our knowledge, this is among the first studies to provide initial indications that Razuprotafib might help reduce ECC-induced systemic and interstitial inflammation. Further molecular studies are required to elucidate the mechanisms by which VE-PTP inhibition functions during ECC. 

Finally, we administered a single dose of Razuprotafib, whereas studies on more chronic conditions, like ocular disorders, have focused on long-term treatment with multiple daily injections over several weeks. The single-dose administration aligned with the focus of our research, which aimed to explore the one-time use of Razuprotafib just prior to the initiation of ECC, given its expected plasma half-life of approximately one hour [17]. Razuprotafib was well tolerated in patients with ocular disorders [17,43], but two phase 2 COVID19 ARDS trials were discontinued due to concerns about drug-induced hypotension [44,45], a major side effect we also observed in our rats. Vascular dysfunction from systemic inflammation has clinically meaningful adverse outcomes in the intensive care unit (ICU). Clinical care for ECMO patients in the ICU still remains largely supportive rather than targeted at the underlying pathophysiological mechanisms that induce multiple organ failure. Unfortunately, Tie2 activation via VE-PTP inhibition seems more complex than activation through angiopoietin-1 [6] and, in our acute setting, failed to improve ECC-induced organ hyperpermeability and microcirculatory perfusion disturbances.

## 4. Materials and Methods

### 4.1. Experimental Set-Up

All procedures were approved by the Institutional Animal Care and Use Committee of the University of Amsterdam, The Netherlands (Animal welfare number: AVD1180020172144), and they were conducted following the EU Directive (2010/63EU) on the protection of vertebrate animals used for experimental and other scientific purposes and the ARRIVE guidelines on animal research [46].

Male *Wistar* rats of 375–425 g (Charles River Laboratories, Brussels, Belgium) were housed in a temperature-controlled room (12/12 h light dark cycle, 20–23 °C, 40–60% humidity) with food and water ad libitum. Rats were randomly assigned to undergo 75 min of venoarterial extracorporeal circulation (ECC) with either Razuprotafib treatment or vehicle as the control (n = 11 per group) in a blinded manner. One hour after discontinuation of ECC, rats were killed (Figure 7). A separate group of rats underwent a sham procedure (control, n = 8). These rats underwent similar surgical preparations but were not connected to the ECC circuit, and they were monitored for two hours and fifteen minutes.

Hemodynamic and microcirculatory perfusion measurements were performed at baseline, 10 and 60 min after the initiation of ECC, and 10 and 60 min after the discontinuation of ECC. Kidneys, lungs, and blood samples were collected and stored at −70 °C for additional molecular analyses.

### 4.2. Anesthesia, Analgesia, and Surgical Preparation

All animals were anesthetized as previously reported with 4.0% isoflurane (Karizoo, Barcelona, Spain) in a plastic box filled with oxygen-enriched air [6,8,22]. Following endotracheal intubation with a 14 G catheter (Venflon Pro, Becton Dickinson, Helsingborg, Sweden), lungs were mechanically ventilated (UMV-03, UNO Roestvaststaal BV, Zevenaar, The Netherlands; PEEP 2–4 cm H_2_O, respiratory rate of 60–80 breaths/min, tidal volume ~10 mL/kg), and anesthesia was maintained with 1.0–2.0% isoflurane in oxygen-enriched air (40% O_2_/60% N_2_). Additionally, fentanyl (12 µg/kg, Janssen-Cilag, Tilburg, The Netherlands) was administered as additional analgesia and repeated approximately every 45 min. Respiratory rate was adjusted based on blood gas values to maintain pH and partial pressure of carbon dioxide within physiological limits. The depth of anesthesia was continuously monitored and adjusted if necessary based on heart rate and mean arterial pressure.

A polythene cannula (ICU Medical, Houten, The Netherlands) was placed in the carotid artery for continuous measurements of arterial blood pressure and blood withdrawal for blood gas analysis and hematocrit measurements (RAPIDPoint 500, Siemens, Munich, Germany). Arterial blood pressure, ECG, and heart rate were continuously recorded using PowerLab software (PowerLab 8/35, Chart 8.0; AD Instruments Pty, Ltd., Castle Hill, Australia).

The left cremaster muscle was isolated under warm saline superfusion, spread out on a heated platform (34 °C), and covered with gas impermeable plastic film (Saran wrap) for cremaster perfusion measurements as previously described [6,8,22,29].

Heparin (500 IU/kg, LEOPharma, Amsterdam, The Netherlands) was administered followed by cannulation of the right jugular vein with a modified cannula (PerkuFlow n. Schlottmann, Pflugbeil GmbH, Zorneding, Germany) for venous inflow of the ECC circuit. The right femoral artery was cannulated with a 20 G catheter (Arterial Cannula, Becton Dickinson, Helsingborg, Sweden) for arterial outflow of the circuit. All catheter insertions were preceded by the local application of 1% lidocaine. Before initiation of the study protocol, an additional dose of heparin (500 IU/kg) was given in combination with rocuronium bromide (1.5 mg/kg, Fresenius Kabi, Halden, Norway).

### 4.3. Treatment with Razuprotazib, a Small-Molecule Inhibitor of VE-PTP

After baseline measurements, but before the initiation of ECC, rats received a single dose of 20 mg/kg Razuprotafib dissolved in 3% dextrose (AKB-9778, HY-109041, MedChemExpress, Monmouth Junction, NJ, USA) or 3% dextrose as the control (vehicle) in a blinded manner via the jugular vein. The dose was determined in consultation with Aerpio Pharmaceuticals, the biotechnology company that originally developed Razuprotafib.

### 4.4. Extracorporeal Circulation

The ECC circuit consists of an open venous reservoir, a roller pump (Pericor SF70, Verder, Haan, Germany), and an oxygenator–heat exchanger with a three-layer hollow fiber membrane for gas exchange (Ing. M. Humbs, Valley, Germany). A 1.0 mm diameter arterial line (LectroCath, Vygon, Ecouen, France) was connected to the femoral outflow catheter. To maintain target ECC flow rates > 140 mL/kg/min, additional doses of albumin were administered when necessary. If necessary, boluses of phenylephrine (10 µg) were administered to maintain a mean arterial pressure above 60 mmHg. The discontinuation of ECC occurred after 75 min by removing the venous cannula and clamping the jugular vein.

### 4.5. Cremaster Microcirculatory Perfusion

Microcirculatory perfusion measurements were performed using a 10× objective on an intravital microscope (AxiotechVario 100HD, Zeiss, Oberkochen, Germany) connected to a digital camera (acA720-520uc, Basler, Ahrensburg, Germany) with a final magnification of 640× [6,8,22,29]. Briefly, three regions of the microvasculature (vessels up to 20 µm diameter) in the cremaster muscle with adequate perfusion quality were selected during the baseline. These exact predefined regions were followed throughout the experiment: after surgical preparation of the cremaster muscle before the start of ECC (baseline), 10 min after the initiation of ECC (10′ ECC), 60 min after the initiation of ECC (60′ ECC), 10 min after the discontinuation of ECC (10′ post-ECC), and 60 min after the discontinuation of ECC (60′ post-ECC) (Figure 7).

For perfusion analyses, two vertical lines were drawn in each video screen. The total number of capillaries per screen was obtained by averaging the counted capillaries crossing the two vertical lines. These small vessels were categorized as continuously perfused, intermittently perfused (blood flow was arrested at least once or flow was reversed), or non-perfused capillaries (vessels without erythrocytes or non-flowing erythrocytes). Finally, the proportion of continuously perfused vessels (PPV) was calculated by the ratio of the absolute number of continuously perfused vessels and the total number of vessels.

### 4.6. Renal and Pulmonary Edema

Kidney and lung tissue was harvested at the end of the experiment under terminal anesthesia. Wet tissue was weighed and dried at 37 °C. After 48 h, dry tissue was weighed, and the wet-to-dry weight ratio was calculated as an estimate for tissue water content.

### 4.7. Vascular Leakage

Vascular leakage was determined by the extravasation of FITC-labeled dextrans [47]. Fifteen minutes before killing the rats, 6.25 mg of FITC-labeled dextrans of 70 kDa (FD70S-1G, Sigma-Aldrich; Saint Louis, MO, USA) was administered intravenously. Fifty milligrams of lung or kidney tissue was homogenized in RIPA buffer with protease and phosphatase inhibitors (cOmpleteTM Protein Inhibitor Cocktail, Roche Diagnostics, Almere, The Netherlands) and centrifuged for 15 min at 13,000× *g* rpm at 4 °C. Fluorescence intensity in lung and kidney homogenates were determined at an excitation wavelength of 485 nm and an emission wavelength of 535 nm using a spectrophotometer (SpectraMax^®^ M2e, Molecular Devices, LLC, San Jose, CA, USA) and were converted to µg/mg tissue using a standard curve.

### 4.8. Plasma Analyses

Arterial blood was collected at the baseline, 60 min after the initiation of ECC (60′ ECC) and 60 min after the discontinuation of ECC (60′ post-ECC). Plasma levels of IL-6, TNFα, and ICAM-1 were measured using a Luminex platform (Biotechne, Dublin, Ireland). Plate-to-plate variation was accounted for by using negative and positive controls. Values below the detection limit were imputed with the lower limit of quantification given by the calibration curve for the univariate comparisons. Measurements were judged to be unreliable when less than 25 beads were counted or extrapolated outside of the reference standard concentrations and were, therefore, excluded for analysis.

Plasma levels of angiopoietin-1 (SEA008Ra, Cloud-Clone Corporation, Katy, TX, USA), angiopoietin-2 (SEA009Ra, Cloud-Clone Corporation, Katy, TX, USA), and soluble Tie2 (MBS-036226, MyBioSource, San Diego, CA, USA) were measured with ELISA in accordance with the manufacturer. To account for changes in the plasma compartment, all plasma marker concentrations were adjusted for hematocrit level fluctuations relative to the baseline. This correction was applied by dividing the measured plasma concentration per unit plasma volume by the hemoconcentration factor h(p) calculated as h(p) = H(x) [100 − H(b)]/(H(b) [100 − H(x)]), where H(b) represents the baseline hematocrit (in percent) and H(x) the hematocrit at any given time point x.

### 4.9. RNA Analyses

Total ribonucleic acid (RNA) was extracted from 10–15 mg of frozen kidney and lung tissue and was isolated using the RNeasy mini kit (Qiagen, Venlo, The Netherlands) [24,48]. RNA concentration and purity were determined using NanoDrop One (NanoDrop Technologies, Wilmington, DE, USA). A total of 1 µg of RNA was transcribed into complementary DNA using an iScript™ cDNA synthesis kit (Bio-Rad, Veenendaal, The Netherlands) using oligo-dT priming. Messenger RNA (mRNA) abundance was determined with the use of assay-on-demand primers/probe sets of the following genes: *Rplp0*, *Tie1*, *Tie2*, and *VE-PTP* (Applied Biosystems, Foster City, CA, USA). The details of all genes and primers are visualized in Table 2. mRNA abundance was measured using a LightCycler^®^ 480 real-time PCR system (Roche Diagnostics, Almere, The Netherlands) and was normalized to the expression of the housekeeping gene *Rplp0*, yielding the ΔCT value.

### 4.10. Histology

Lung and kidney tissue was fixed in 4% formalin and embedded in paraffin. Four-micrometer-thick paraffin sections were stained with hematoxylin and eosin. Sections were scored on a scale from 0 to 4 by a pathologist who was blinded for treatment allocation [47]. Renal sections were scored for interstitial edema and tubular injury. Tubular injury was characterized by necrosis, dilation, cast deposition, and loss of brush border. Pulmonary sections were scored for hemorrhage, interstitial inflammation, neutrophil sequestration, perivascular edema, and alveolar edema. A score of 0 represented normal organs; 1 represented mild, less than 25% involvement; 2 represented moderate, 25 to 50% involvement; 3 represented severe, 50 to 75% involvement; and 4 represented very severe, more than 75% involvement.

### 4.11. Statistical Analysis

Sample size was calculated based on the assumption that treatment with Razuprotafib is as equally effective as the angiopoietin-1 mimetic, Vasculotide, which preserved microcirculatory perfusion in rats on cardiopulmonary bypass (CPB + Vasculotide vs. pre-CPB: 8.8 vs. 10.9 vessels per recording) compared to untreated rats on CPB (CPB + placebo vs. pre-CPB: 5.4 vs. 9.9 vessels per recording) [6]. Using an alpha of 0.05, a power of 0.90, and a standard deviation of 1.1, a sample size of 11 rats per group was calculated using a two-way analysis of variance (ANOVA) test design using nQuery 8.5 (Statsols, Boston, MA, USA). Rats that dropped out during the experiment were replaced.

Data are expressed as median [interquartile range] or mean ± standard deviation and were analyzed using GraphPad Prism 10.2 (GraphPad Software, Boston, MA, USA). The normality of distribution was tested with the Shapiro–Wilk test. One-way ANOVA with Bonferroni post hoc analysis or the Kruskal–Wallis test with Dunn’s post hoc analyses were used to evaluate differences between groups for normally and not normally distributed data, respectively. Time-dependent differences were analyzed with two-way ANOVA with repeated measures and Bonferroni post hoc analysis. *p* values < 0.05 were considered statistically significant.

## 5. Conclusions

In conclusion, ECC is associated with systemic inflammation, endothelial damage, a dysbalanced endothelial angiopoietin/Tie2 system, renal edema, disturbed microcirculatory perfusion, and impaired respiratory function. Treatment with the VE-PTP inhibitor Razuprotafib did not improve ECC-induced microcirculatory perfusion disturbances nor renal edema. Future studies should validate the mechanism of action of Razuprotafib through direct VE-PTP, Tie2, and VE-cadherin phosphorylation assays. These findings provide valuable insights into endothelial-targeted interventions during ECC and highlight the need for further research to refine strategies for reducing organ failure in ECMO patients using our translational rat model.

## Figures and Tables

**Figure 1 ijms-26-03000-f001:**
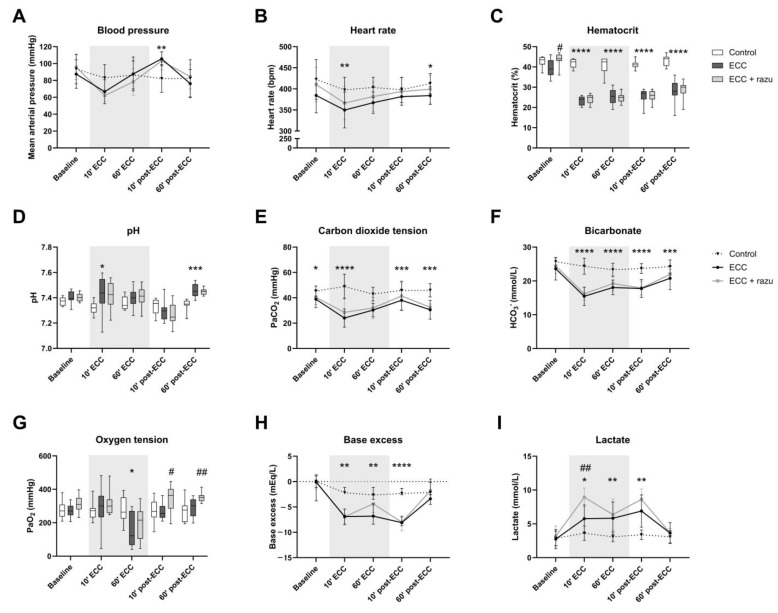
Hemodynamics and arterial blood gas analysis. Mean arterial pressure (**A**), heart rate (**B**), hematocrit (**C**), pH (**D**), carbon dioxide tension (PaCO_2_, **E**), bicarbonate (HCO_3_^−^, **F**), oxygen tension (PaO_2_, **G**), base excess (**H**), and lactate (**I**) in rats during and after extracorporeal circulation (ECC; continuous black line or box, n = 11), ECC with Razuprotafib treatment (ECC + razu, continuous grey line or box, n = 11), or sham rats (control; dashed line or white box, n = 8). The gray background indicates the ECC period. Data are presented as mean ± standard deviation (lines) or median with interquartile and full range (boxes), * *p* < 0.05, ** *p* < 0.01, *** *p* < 0.001, **** *p* < 0.0001 ECC effect, # *p* < 0.05, ## *p* < 0.01 Razuprotafib effect.

**Figure 2 ijms-26-03000-f002:**
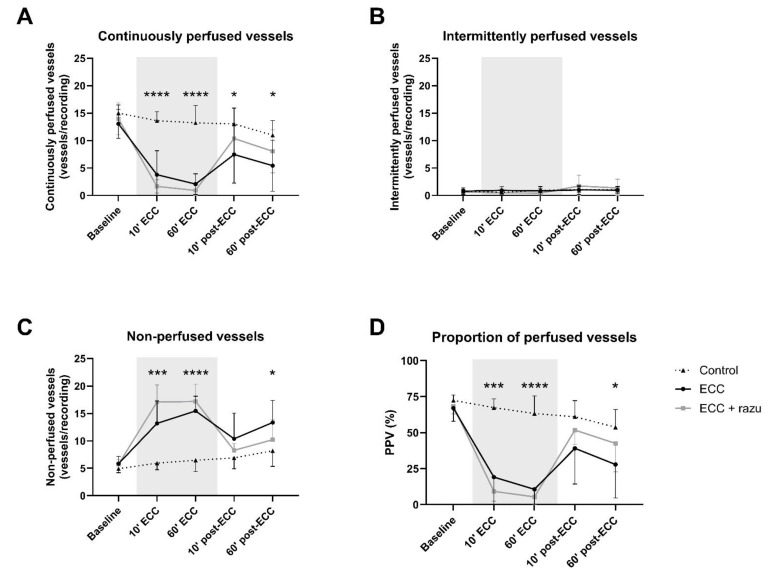
Microcirculatory perfusion. Continuously perfused vessels (**A**), intermittently perfused vessels (**B**), non-perfused vessels (**C**), and the proportion of perfused vessels (PPV; **D**) in cremaster muscle in rats during and after extracorporeal circulation (ECC; continuous black line, n = 11), ECC with Razuprotafib treatment (ECC + razu, continuous grey line, n = 11), or sham rats (control; dashed line, n = 8). The gray background indicates the ECC period. Data are presented as mean ± standard deviation, * *p* < 0.05, ****p* < 0.001, **** *p* < 0.0001 ECC effect.

**Figure 3 ijms-26-03000-f003:**
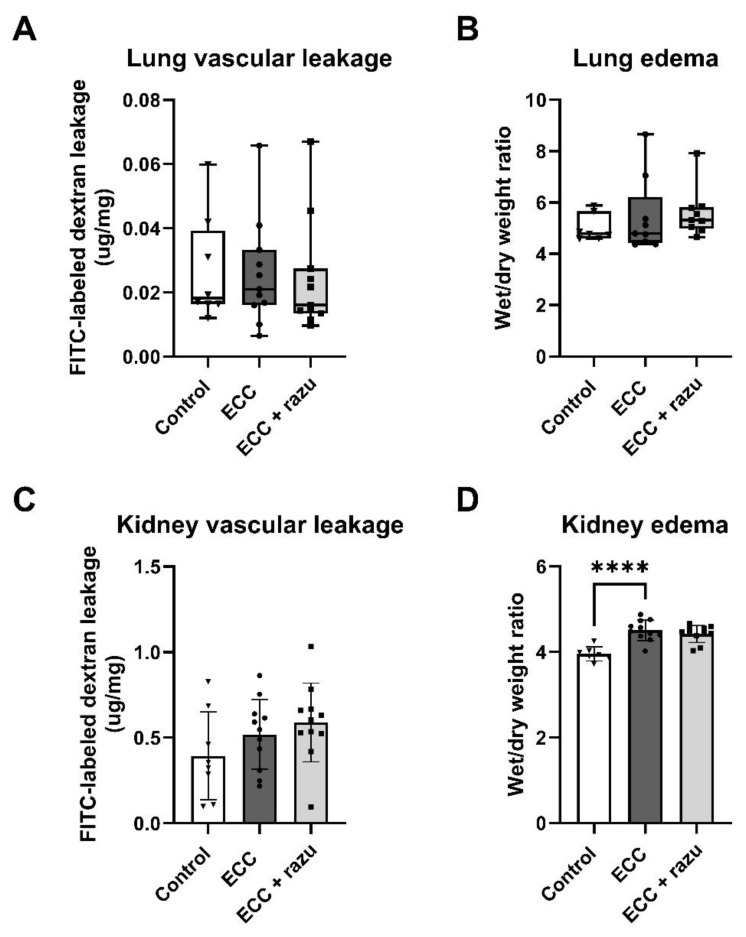
Organ permeability and fluid requirements. Pulmonary FITC-labeled dextran extravasation (**A**), pulmonary wet-to-dry weight ratio (**B**), renal FITC-labeled dextran extravasation (**C**), and renal wet/dry weight ratio (**D**) in rats with extracorporeal circulation (ECC), ECC with Razuprotafib treatment (ECC + razu), or sham rats (control). Each dot represents an individual rat. Data are presented as mean ± standard deviation (bars) or median with interquartile and full range (boxes). **** *p* < 0.0001 ECC effect.

**Figure 4 ijms-26-03000-f004:**
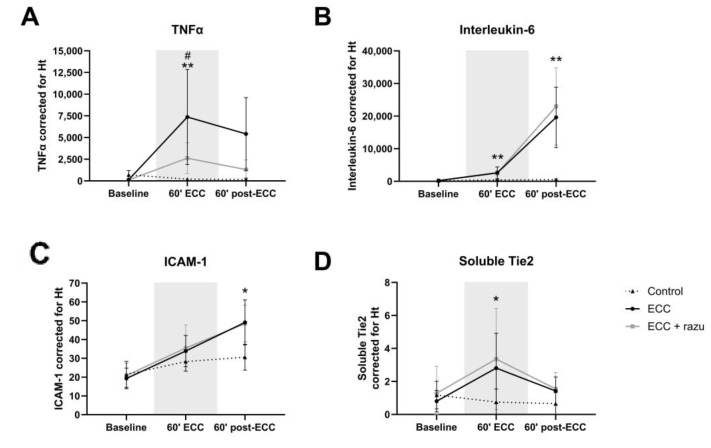
Circulating markers of inflammation and the angiopoietin/Tie2 system. Plasma levels of TNFα (**A**), interleukin-6 (**B**), ICAM-1 (**C**), soluble Tie2 (**D**), angiopoietin-2 (**E**), and angiopoietin-1 (**F**) in rats during and after extracorporeal circulation (ECC; continuous black line or box, n = 11), ECC with Razuprotafib treatment (ECC + razu, continuous grey line or box, n = 11), or sham rats (control; dashed line or white box, n = 8). The gray background indicates the ECC period. Data are presented as mean ± standard deviation (lines) or median with interquartile and full range (boxes), * *p* < 0.05, ** *p* < 0.01, *** *p* < 0.001. ECC effect, # *p* < 0.05 Razuprotafib effect.

**Figure 5 ijms-26-03000-f005:**
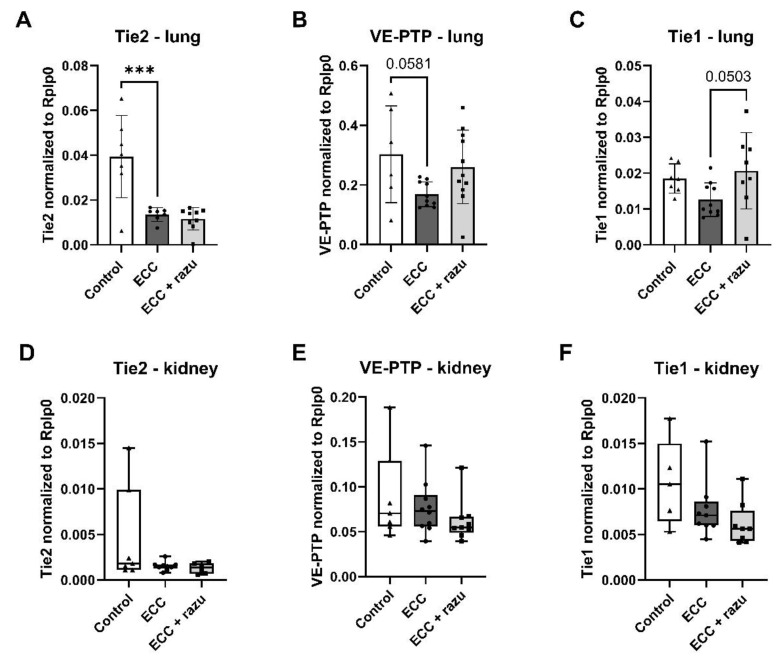
Gene expression of endothelial receptors related to the angiopoietin/Tie2 system. mRNA levels of *Tie2* (**A**,**D**), *VE-PTP* (**B**,**E**), and *Tie1* (**C**,**F**) in the lungs (**A**–**C**) and kidneys (**D**–**F**) of rats after extracorporeal circulation (ECC, n = 11), ECC with Razuprotafib treatment (ECC + razu, n = 11), and sham rats (control, n = 8). Each dot represents an individual rat. Data are presented as mean ± standard deviation (bars) or median with interquartile and full range (boxes), *** *p* < 0.001 ECC effect.

**Figure 6 ijms-26-03000-f006:**
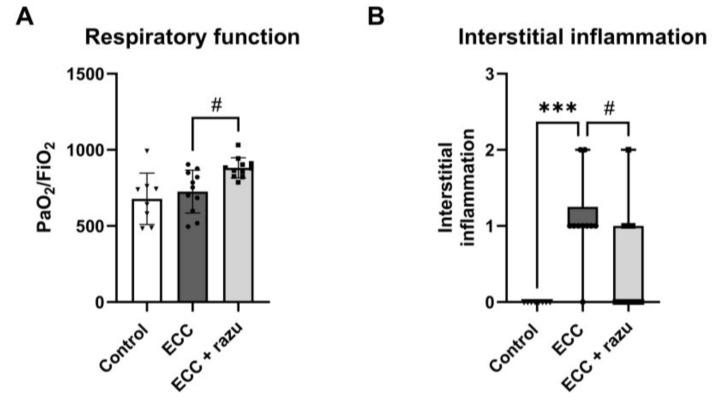
Razuprotafib reduced ECC-induced lung injury. PaO_2_/FiO_2_ ratio (**A**) and pulmonary interstitial inflammation (**B**) in rats 60 min after weaning from extracorporeal circulation (ECC), ECC with Razuprotafib treatment (ECC + razu) or sham procedure (control). Data are presented as mean ± standard deviation (bars) or median with interquartile and full range (boxes). Each dot represents an individual rat. *** *p* < 0.001 ECC effect, # *p* < 0.05 Razuprotafib effect.

**Figure 7 ijms-26-03000-f007:**
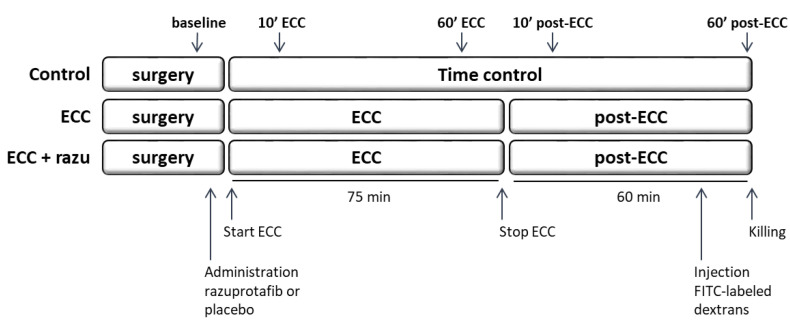
Experimental set-up. Rats were divided into three groups: extracorporeal circulation with placebo (ECC, n = 11), ECC with Razuprotafib treatment (ECC + razu, n = 11), and sham surgery (control, n = 8). Razuprotafib or the placebo was administered directly after baseline measurements followed by 75 min of ECC in the ‘ECC’ groups. One hour after the discontinuation of ECC and 15 min after the administration of FITC-labeled dextrans, rats were killed and kidney and lung tissue was harvested. Microcirculatory perfusion measurements were performed at baseline, 10 and 60 min after initiation of ECC (10′ and 60′ ECC), and 10 and 60 min after stopping ECC (10′ and 60′ post-ECC).

**Table 1 ijms-26-03000-t001:** Kidney and lung histopathology.

	Control (n = 8)	ECC (n = 11)	ECC + Razuprotafib (n = 11)
Kidney			
Interstitial edema	0 [0–0]	0 [0–0]	0 [0–0]
Tubular injury	0 [0–0]	0 [0–0]	0 [0–0]
Lung			
Perivascular edema	2 [1–3]	2 [2–3]	2 [2–3]
Alveolar edema	0 [0–1]	0 [0–0]	0 [0–0]
Neutrophil sequestration	1 [0–1]	1 [0–1]	1 [1–1]
Interstitial inflammation	0 [0–0]	1 [1–1] ***	0 [0–1] #
Hemorrhage	0 [0–0]	0 [0–0]	0 [0–0]
Thrombi	0 [0–0]	0 [0–0]	0 [0–0]

Data are presented as median [IQR] based on a score from 0 to 4. A score of 0 represented normal lungs or kidneys; 1 represented mild, less than 25% involvement; 2 represented moderate, 25 to 50% involvement; 3 represented severe, 50 to 75% involvement; and 4 represented very severe, more than 75% involvement. Data are presented as median with interquartile and full range. *** *p* < 0.001 ECC effect, # *p* < 0.05 Razuprotafib effect.

**Table 2 ijms-26-03000-t002:** Real-time qPCR primers.

Gene	Assay ID	Encoded Protein
*Tek*	Rn01433337_m1	Endothelial-specific receptor tyrosine kinase (Tie2)
*Tie1*	Rn01417182_m1	Tyrosine kinase with immunoglobulin-like and EGF-like domains 1 (Tie1)
*Ptprb*	Rn01502973_m1	Protein tyrosine phosphatase, receptor type, B (VE-PTP)
*Rplp0*	Rn00821065_g1	60S acidic ribosomal protein P0 (Arbp)

VE-PTP, vascular endothelial-protein tyrosine phosphatase.

## Data Availability

The experimental data that support the findings of this study are available in DANS Data Station Life Sciences with the identifier https://doi.org/10.17026/LS/QEGE6A [49].

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
