# Peer review of "Razuprotafib Does Not Improve Microcirculatory Perfusion Disturbances nor Renal Edema in Rats on Extracorporeal Circulation"

_ijms, 2025, doi:10.3390/ijms26073000_

Round 1

Reviewer 1 Report

Comments and Suggestions for Authors

The authors present a study to test whether razuprotafib can mitigate extracorporeal circulation-induced ALI in a rat model. The ECMO model is well characterized, but there are significant concerns regarding whether it has been deployed successfully given the hemodynamic instability that occurs quickly in the ECC animals. In terms of the effects of razuprotafib on ALI mitigation, the ALI that does develop appears to be quite subtle, and it is not clear that the drug truly reduces it in a meaningful way. The drug is reported to impact a few analytes here and there (TNFa at 60 min, per se), but a convincing story that supports that the drug works to protect organs is missing. If this is really a negative study, then it should be framed as such. Even the title appears to overstate the study as there was very limited lung injury seen. A number of other issues should be addressed in a revision, described below:

  1. The use of a rat model of ALI could confound applicability to other mammal species, as rats of an endothelial injury dominant pattern of ALI, whereas other mammals (mice, primates) it is more epithelial dominant. As the authors are studying primarily markers of endothelial injury, this model makes sense, though it may not be as readily validated in other species. This should be stated as a limitation.
  1. The experimental design is really a 2x2 factorial design but is missing a Razu only control group. This should be added.
  1. As it is currently written, the abstract results section is hard to read and stay organized on what is up, what is down, and in which organ. It would be helpful to organize the information to tell more of a story. “VE-PTP” should be defined upon first use.
  1. Please explicitly define “microcirculatory perfusion disturbances” (does this mean capillary flow?) and “organ hyperpermeability” (does this mean edema?). Are these terms that have been validated in other studies for the purposes studied here?
  1. The resolution of Figure 1 is not sufficient to clearly see what is an asterisk vs. what is a hashtag. Was lactate higher at 10 min in the ECC + razu group? For O2 tension at 60’ post-ECC? It is hard to tell.
  1. There are concerns that the ECC procedure itself caused significant organ hypoperfusion so quickly, within 10 minutes of initiation, given the increased lactate and decrease microcirculatory perfusion at this time point. This would suggest the flows were too low, or there was substantial blood loss from the cannulations (for instance, Hct dropped by ~50% within 10 minutes? That shouldn’t happen outside of profound blood loss). Observing these findings at 10 minutes seems more likely due to issues with the model rather than endothelial injury, per se.
  1. The methods section states that RR was adjusted to maintain pCO2 and pH within physiological limits, although both parameters were significantly different between groups. This adds a potential confounder to the study interpretation.
  1. Suggest moving the conclusion paragraph from after the methods to after the discussion.
  1. More details of the sham procedure should be provided. What exactly did it consist of?
  1. If histopathological assessment did not reveal interstitial edema in the kidney, what explains the increased wet-to-dry weight? Higher urine content?
  1. It is unexpected that the authors would correct the plasma concentrations of TNFa, IL-6, ICAM-1, etc. for hematocrit, and sufficient details are not provided as to exactly how this correction was made. These molecules would be found in the plasma fraction of whole blood, not the RBC fraction. Adjustment would need to take into account the hemoconcentration (or expansion) of the plasma compartment, not the blood compartment (see PMID: 22370687 for guidance on this correction).
  1. It is not clear what the significance of higher Ang-2 at 60’ min post-ECC in the ECC + razu group, but this would be concerning for more vascular injury, no?
  1. Figure 5 – what is Rplp0 and why was gene expression normalized to this?
  1. Figure 6 is missing from the manuscript.
  1. I don’t think it is quite fair to say that treatment with Razu “suppressed the inflammatory response”. There was a single reduction in TFN-alpha at 60 min, however, IL-6 was not suppressed, and other markers were not tested. Furthermore, there was no decrease in lung neutrophil sequestration. The decrease in “interstitial inflammation” should be further described – where these also neutrophils? It seems like a lost opportunity not to also report cytokine levels in the lung and kidney tissue itself. Surely the authors collected it.

Reviewer 2 Report

Comments and Suggestions for Authors

The manuscript presented for review titled “Razuprotafib mitigates extracorporeal circulation-induced lung injury in rats without affecting organ hyperpermeability or microcirculatory perfusion disturbances" investigate whether Razuprotafib, a vascular endothelial protein tyrosine phosphatase (VE-PTP) inhibitor, can mitigate ECC-induced lung injury in rats. Blocking VE-PTP is expected to boost Tie2 activity, strengthening blood vessel integrity and enhancing microcirculatory flow.
The subject matter of this article is important as it enhances our insight into endothelial dysfunction in ECC and explores possibilities for its therapeutic intervention.
The language of the work is understandable and easy to read. 
However, certain areas of the manuscript require major revisions to improve the clarity and coherence of the manuscript.

1. There is no clear mechanistic explanation for why Razuprotafib failed to enhance microcirculatory perfusion or reduce organ permeability, despite its anticipated impact on Tie2 activation and endothelial stability. The study proposes that inhibiting VE-PTP would boost Tie2 activation, thereby strengthening endothelial junctions. However, since Tie2 phosphorylation (p-Tie2), the primary indicator of activation, is not measured, there is no direct proof of Tie2 activation. Consequently, it remains uncertain whether Razuprotafib successfully targeted its intended mechanism.
2. An unexpected discovery was that Razuprotafib led to an additional rise in circulating Angiopoietin-2 levels following ECC. The authors should explore potential mechanisms by which Razuprotafib may contribute to the upregulation of Angiopoietin-2.
3. This study administrers a single dose of Razuprotafib, whereas most prior research on Tie2 activation involved chronic dosing regimens. At a minimum, the authors should recognize this limitation and discuss how prolonged treatment could produce different outcomes, even if a chronic administration model is not feasible.

Round 2

Reviewer 1 Report

Comments and Suggestions for Authors

While the authors have addressed some of the concerns raised, a number have gone unaddressed. There is not enough consistent data to support that the drug may be beneficial and some data to suggest it is harmful (higher lactate). This alone is cause for more rat experiments, not less as the authors suggest, before this goes to human subjects. The lung injury is too subtle to have confidence that the drug reduces inflammation. The correction for the plasma fraction of blood volume rather than hematocrit has perhaps been misunderstood by the authors and details of the equation used to make the correction are still lacking. Furthermore, the effect on the hematocrit by the ECC but not the sham is a concern and raises question of whether the sham procedure is a sufficient control.

Reviewer 2 Report

Comments and Suggestions for Authors

I appreciate the efforts made by the authors in revising the manuscript. The authors have made significant efforts to improve the clarity and coherence of the manuscript based on reviewer feedback.

Although the discussion now recognizes the lack of direct Tie2 phosphorylation (p-Tie2) measurements, this remains a significant limitation in understanding whether Razuprotafib actually achieved its intended mechanism of action.

Suggestion is that the authors should clearly state in the conclusion that future studies need to validate Razuprotafib’s mechanism through Tie2 phosphorylation assays. This would make their recommendation for mechanistic validation more explicit.

Round 3

Reviewer 1 Report

Comments and Suggestions for Authors

I thank the authors for their attention to my concerns. The manuscript has been improved in the last round of revision. After re-reading it, and considering the fact that there is very little lung injury to speak of, I think the purported benefits to the lung that are discussed in the Discussion section starting in line 302 should still be toned down. Furthermore, I have been pondering the reported (and perplexing) improvement in P/F ratio in the Razu group. First, I am not sure that P/F ratio can really be a valid outcome if the ECC is adding extracorporeal oxygen to the blood stream and bypassing the lung. Is this the case? If so, then it confounds any P/F ratio as an additional unmeasured variable. That should be mentioned as a mitigating factor when drawing conclusions. Second, because we know that Razu causes hypotension and increased lactate, indicative of increased anaerobic metabolism, it seems to me that a possible (or even likely) explanation for the higher P/F ratios in the Razu group is simply due to reduced tissue O2 uptake, leading to higher PaO2 levels, and falsely elevating P/F ratios. This should also be mentioned as something that cannot be ruled out in the Discussion section.
